# Performance of Autumn and Spring Calving Holstein Dairy Cows with Different Levels of Environmental Exposure and Feeding Strategies

**DOI:** 10.3390/ani13071211

**Published:** 2023-03-31

**Authors:** Maria Noel Méndez, Lucía Grille, Graciana R. Mendina, Peter H. Robinson, María de Lourdes Adrien, Ana Meikle, Pablo Chilibroste

**Affiliations:** 1Departamento de Producción Animal y Pasturas, Facultad de Agronomía, Universidad de la República, Paysandú 60000, Uruguay; 2Departamento de Ciencias Veterinarias y Agrarias, Facultad de Veterinaria, Universidad de la República, Paysandú 60000, Uruguay; 3Department of Animal Science, University of California, Davis, CA 95616, USA; 4Laboratorio de Endocrinología y Metabolismo Animal, Facultad de Veterinaria, Universidad de la República, Montevideo 13000, Uruguay

**Keywords:** grazing, mixed ration, confinement, heat stress, heavy rain, full lactation performance

## Abstract

**Simple Summary:**

Since there has been an increase in the frequency of extreme weather events (i.e., heavy rains, heat stress) due to climate change, the interaction between feeding and management issues and the required facilities to alleviate environmental effects on animal performance has become relevant. Although there is extensive literature on the implementation of confined vs. mixed (grazing + mixed ration) feeding strategies in dairy systems, most are short-term studies and there is little information on their effects on cow full lactation performance, associated with environmental exposure stress throughout their productive cycle, which depends on their calving season. This manuscript determines and interprets its effects on milk production and composition and energy balance (i.e., body condition score, non-esterified fatty acids, and beta-hydroxybutyrate) during a full lactation in two calving seasons, addressing whole-herd (extensive to whole-farm) feeding and management issues. The results demonstrate that outdoor soil-bedded milk production systems, when well-managed, have a very high milk production potential that could equate to the productive response of improved infrastructure systems (i.e., a compost-bedded pack barn with cooling capacity) under moderately unfavorable environmental conditions (i.e., infrequent heavy rains), but in worse situations (i.e., severe heat waves and frequent heavy rains), performance could be compromised.

**Abstract:**

Environmental exposure during confinement and feeding strategy affects cow behavior, nutrient utilization, and performance. Milk production and composition, body condition score, non-esterified fatty acids, and beta-hydroxybutyrate were determined during a full lactation in cows submitted to (a) grazing + partial confinement in outdoor soil-bedded pens with shade structures (OD-GRZ); (b) grazing + partial confinement in a compost-bedded pack barn with cooling capacity (CB-GRZ); or (c) total confinement (same facilities as CB-GRZ) and fed TMR ad libitum (CB-TMR). Autumn (ACS) and spring (SCS) calving season cows were used for each treatment, except for CB-TMR (only SCS). In ACS, treatments did not differ in any variable, possibly due to mild weather. In SCS, milk production was higher in CB-TMR than CB-GRZ, which in turn produced more milk than OD-GRZ. Differences coincided with heat waves and/or heavy rains (similar grazing conditions and mixed ration DM intake). Milk fat, protein and lactose yield, protein content, and BCS were higher in CB-TMR, without differences between CB-GRZ and OD-GRZ. Cows in OD-GRZ had impaired energy metabolism. Under moderately unfavorable environmental conditions (ACS), when well-managed, OD-GRZ systems could equate to the productive response of CB-GRZ. However, in worse climatic conditions (SCS), performance could be compromised, especially when compared to TMR systems.

## 1. Introduction

Cows in totally confined milk production systems fed total mixed rations (TMR) have higher intake, production, and energy balance than those in pastoral systems that harvest their forage. Nevertheless, mixed systems (grazing + a MR; MS) can capture some benefits of confined systems while maintaining relatively low feeding costs, which could improve the economics of totally confined systems [1,2,3].

Intensification of pasture-based milk production systems through stocking rate increments results in higher inclusion of supplements in the diet. Indeed, use of conserved forage in conjunction with concentrates in a MR results in higher dry matter (DM) intake and higher milk and solids production [1,2]. This production strategy involves cows being out of the paddock 40 to 60% of the time in an area where feed supplements are provided [3]. Furthermore, high pasture grazing intensity associated with high stocking rates leads to longer periods without access to pasture in order to recover forage in the system, thereby generating the need for facilities to confine the cows [4].

Calving distribution through the year has important effects on the annual milk re-mittance pattern [5] and, consequently, on milk industry supply. Autumn/winter calved cows achieve more total milk production than those calved in spring. They also have longer lactation and a shorter calving to conception interval [6]. However, in Uruguayan winters, a lower pasture growth rate and heavy rains can prevent pasture access, resulting in less directly harvested forage inclusion in the diet of autumn calving cows (ACS) during early lactation. This results in higher feed supplementation and more time in confinement facilities. Although spring calving cows (SCS) achieve higher pasture DM intake and lower production costs, cows experience the challenge of maintaining normothermia in addition to the stress of early lactation [7], which can compromise welfare and productivity. Each dairy system must be evaluated for the opportunities and weaknesses of each calving strategy and adopt the one that best suits its productive objectives.

When confinement time becomes considerable, the negative effects of exposure to environmental conditions on cow comfort, behavior, and performance are accentuated [8]. During heat stress (HS), energy demand for the thermoregulation of immune system hyperactivation diverts the energy supply to the mammary gland [9,10]. This, together with reduced nutrient absorption and lower udder nutrient uptake [11,12], causes milk production decreases of up to 35% [10,13]. On the other hand, under adverse winter conditions, with cold and mud in feeding and resting areas, cows spend more time standing and less time lying [14,15]. Lying time is critical not only because it is associated with rumination time [16], but it is also important for cows to achieve a sufficient amount of quality rest time [17]. Lying deprivation during confinement periods has an impact on behavior during next period activity, which in MS corresponds to grazing [15,16,18]. In addition, cow resistance to moving through muddy areas can cause cows to eat less overall, with fewer larger meals, which negatively impacts ruminal fermentation [19]. As a consequence, nutrient utilization and performance are impaired.

There is scarce literature on pasture-based cow performance under different housing conditions and environmental exposure stress (i.e., cold, mud, or heat stress) at different stages of lactation, depending on calving season strategy. Research on the effects of expo-sure to the environment on intensive grazing systems is not only original but also highly relevant from an economic and environmental perspective for the intensification of dairy systems in the southern hemisphere [4]. Although the implementation of confined vs. MS feeding strategies has research antecedents, the existent literature refers to short-term studies [1,2,20,21,22,23,24], and according to our knowledge, no research has been conducted throughout the productive cycle. Furthermore, compost-bedded pack barns are novel facilities to house animals that allow cows freedom to move and a soft place to lye, thus improving animal welfare while enabling manure recycling, consequently diminishing environmental pollution and productive costs [25,26].

The objectives of this study were to: (a) measure and evaluate the effect of different levels of environmental exposure on the performance of cows consuming MR + grazing in two different strategies of calving season (autumn and spring); (b) compare high and low environmental exposure MS with a 100% confined TMR system when detrimental heat stress effects are most pronounced (i.e., SCS strategy). It was hypothesized that milk cows partially confined in a compost-bedded pack barn with a cooling capacity would improve full lactation milk and solids production and energy balance when compared to cows partially confined in outdoor soil-bedded pens with shade structures. Further, a totally confined system would improve milk and solids production, as well as energy balance, compared to MS-fed cows, obtaining more contrasting responses when compared to cows housed in outdoor soil-bedded pens than when compared to cows housed in a compost-bedded pack barn. 

## 2. Materials and Methods 

### 2.1. Cows and Experimental Design

A total of 80 Holstein cows (2.8 ± 1.25 lactations, 640 ± 85 kg body weight; BW) were used in 2 calving experiments, 1 in each calving season, conducted at the Estación Experimental Dr. M. A. Cassinoni (EEMAC) of the Facultad de Agronomía (Paysandú, Uruguay) of the Universidad de la República (UdelaR). Experimental periods lasted from March 2019 to January 2020 for ACS cows and from August 2019 to May 2020 for SCS cows. The experimental protocol was evaluated and approved by the Comisión Honoraria de Experimentación Animal (CHEA) from UdelaR (Montevideo, Uruguay). All cows were managed equally during the dry and prepartum periods and confined and fed a pre-partum TMR for ~3 weeks before expected calving dates. Autumn-calving cows had calving dates of 18 March 2019 ± 14.5 days and spring-calving cows had calving dates of 16 August 2019 ± 8.2 days. Cows were blocked by BW, lactation number, pre-calving body condition score (BCS) and expected calving date, randomly assigned to treatment, and grouped in 4 pens of 4 cows each (i.e., 16 cows/treatment).

Treatments consisted of: (a) ACS and SCS cows subjected to 8 h in a grazing paddock + supplemental MR in outdoor soil-bedded pens with shade structures during confinement (OD-GRZ; high environmental exposure MS); (b) ACS and SCS cows subjected to 8 h in a grazing paddock + supplemental MR in a compost-bedded pack barn with cooling capacity during confinement (CB-GRZ; low environmental exposure MS); (c) SCS cows subjected to a totally confined system with cows in the same facilities as CB-GRZ but fed a TMR twice daily ad libitum (CB-TMR). 

### 2.2. Management and Feeding 

The CB-TMR and CB-GRZ cows were confined in a fully roofed, compost-bedded pack barn divided into pens, each containing four cows. Compost-bedded pack area was of 13.5 m^2^ per cow and was continued by a concrete area of 6.7 m^2^ per cow with access to a feed bunk per pen with a length of 0.75 m per cow and automatic drinkers to ensure ad libitum access to water. A 15 cm layer of new, fresh bedding material (i.e., rice husks and wood chips) was supplied every 15–20 days. Compost was superficially labored twice a day with a chisel plough in order to remove water vapor, allow oxygen entry, and maintain small, homogeneous particles. The concrete area was cleaned three times a week by trawling with a tractor carrying rubber. The barn had cooling capacity, with continuous-operation fans and sprinklers with automatic operation over 25 °C. 

The OD-GRZ cows were confined in outdoor soil-bedded pens made up of 2 paddocks, alternately occupied according to soil moisture and surface deterioration, with an area of 48 m^2^ per cow. Paddocks had a slight slope for water and manure runoff. Each pen had shade structures of 4.8 m^2^ per cow (nylon roofed at 4.5 m height with a slope of 15%), close to automatic drinkers (same as previous). Feeders were located at the other end of the paddocks, with a length of 1.10 m per cow and a feeding area of 10 m^2^ per cow.

The milk parlor was built 100 m from pens in order to minimize cow activity and long waiting periods during milking.

The TMR/MR were the same for all treatments. It varied over time according to available conserved forage as well as market-available grains and by-products for the commercial concentrate, with a total of 7 combinations used in the 15 months of the experiments (Table 1). Diets were formulated based on the recommendations of the National Research Council (NRC [27]) for 620 kg cows producing 45 L/d of 4% fat-corrected milk. In MS, both pasture and MR were considered nutritionally balanced diets, with MR used as a pasture complement (limited amounts) to achieve the desired DM intake, which was dependent on current pasture stock of the system. 

The MS (i.e., OD-GRZ and CB-GRZ) were high stocking rate systems (i.e., 2.5 lactating cows and/or 1550 kg BW/ha grazing platform), where cows had 8 h of daily access to weekly grazing plots, if allowed by weather conditions and/or grazing platform available herbage mass (HM). Both grazing treatments accessed different grazing plots (all pens of the same treatment together) with similar herbage allowances (HA). From March to October 2019, cows grazed between 7 a.m. and 2 p.m., while from November 2019 to April 2020, cows grazed from 6 p.m. to 2 a.m., in order to minimize heat stress and its negative impact during grazing. The pastures used were: tall fescue (*Festuca arundinacea*), lucerne (*Medicago sativa*) + orchard grass (*Dactylis glomerata*), oat (*Avena sativa*), annual raygrass (*Lolium multiflorum*), and soybean (*Glycine max*). Forage management and chemical composition are summarized by season in Table 2. Herbage mass was determined weekly using the double sampling technique [28] and then calculated pasture growth in order to adjust HA, taking into account sward condition (i.e., number of leaves or axillary buds) and available HM in the total grazing platform. Supplementation with TMR/MR was adjusted to ensure cow requirements and productivity goals were met and to achieve the appropriate pasture rotation length depending on pasture growth rate. 

### 2.3. Data Collection, Measurements and Estimates

Climatic conditions (i.e., ambient temperature, relative humidity, rain) records were obtained from the meteorological agency of the experimental station. Heat stress was calculated using a temperature humidity index (THI) as: (1.8 × ET + 32) − (0.55 − 0.55 × RH/100) × (1.8 × ET − 26.8), where ET is environmental temperature and RH is relative humidity [29]. It was considered a mild heat wave when 2 of the 3 following criteria occurred at least 3 days in a row: daily THI average > 72, maximum daily temperature was >32 °C and/or minimum daily temperature was >23 °C. When all three conditions oc-curred simultaneously, it was considered a severe heat wave [30]. 

Cows were milked at 4 a.m. and 5 p.m. during spring/summer and at 3 a.m. and 4 p.m. during autumn/winter in order to minimize heat stress and its negative impact during grazing, as previously mentioned. Individual cow production was recorded at each milking. Milk samples were collected weekly from calving to 90 days in milk (DIM), biweekly from 91 to 180 DIM, and then monthly to the end of the lactation to determine milk fat, protein, and lactose levels (MilkoScan FossElectric FT2^®^, Drachten, The Netherlands).

The offered and refused TMR/MR were measured weekly, as well as sampled, weighed, and oven-dried at 55 °C for 48 h to calculate dry matter intake. Samples were also analyzed for crude protein (CP), neutral detergent fiber (NDF), and acid detergent fiber (ADF), according to AOAC [31]. Total N for CP estimation used the Kjeldahl method of AOAC [32], which involves sulfuric acid digestion with subsequent distillation and titration. NDF used α-amylase, and, as for ADF, an ANKOM200 Fiber Analyzer (ANKOM Tech. Corp., Fairport, NY, USA) was used. Pasture was also sampled and chemically analyzed weekly.

Daily pasture DM intake (kg DM/cow) was estimated by energy balance according to NRC [27], as the kg of pasture necessary to provide the remaining energy to achieve cow net energy (NE) requirements did not come from MR. Cow NE requirements were estimated as the sum of maintenance, pregnancy, and milk production requirements, taking into account energy contributed or required from losing or gaining BW and BCS [29]. Maintenance requirements were calculated as 80 kcal of NE/kg BW0.75, with a 20% increase for grazing activity in mixed systems [27]. Cow BW was measured monthly for this purpose. The amount of NE/kg BW was calculated according to actual BCS, adjusted for when it was used to support milk production or body deposition [29]. The BCS was assessed biweekly until 120 DIM and monthly from 120 to 305 DIM based on a 5-point scale, according to [33]. Pregnancy requirements were calculated from 190 to 279 days of gestation as: NEL (Mcal/day) = (0.00318 × D − 0.0352) × (CBW/45)/0.218, where D is day of gestation and CBW is calf birth BW in kilograms. Milk NEL concentration for productive requirements (i.e., energy in milk) was calculated from milk production and composition as milk NEL (Mcal/kg) = 0.0929 × Fat% + 0.0547 × CP% + 0.0395 × Lactose%. The NEL provided by pasture and TMR was calculated according to NRC [34] as Pasture NEL (Mcal/kg) = 2.149 − (0.0223 × ADF) and TMR NEL (Mcal/kg) = 1.909 − (0.017 × ADF).

Blood samples were collected biweekly until 120 DIM and monthly from day 120 to 210 DIM, by venipuncture of the coccygeal vein, using 10 mL Vacutest® tubes (Vacutest Kima, Arzergrande, Italy) with heparin. Refrigerated samples were centrifuged at 3000× *g* for 15 min, and plasma was stored at −20 °C until analysis. Non-esterified fatty acids (NEFA) and beta hydroxybutyrate (βHB) concentrations were determined spectrophoto-metrically using commercial kits (Wako NEFA-HR (2) from Wako Pure Chemical Indus-tries Ltd., Osaka, Japan, and Oxidase/Peroxidase, UREA/BUN-UV, Ureasa/Glutamate De-hydrogenase from BioSystems SA, Barcelona, Spain, respectively).

**Table 1 animals-13-01211-t001:** Composition, chemical analysis, and nutritional value of mixed diets fed to lactating cows (% dry matter).

		From		February-19	June-19	July-19	August-19	October-19	March-20	April-20
		To		May-19		August-19	October-19	February-20	April-20	May-20
Experiment				ACS ^1^	ACS ^1^	ACS ^1^	ACS ^1^	ACS ^1^		
							SCS ^2^	SCS ^2^	SCS ^2^	SCS ^2^
Ingredient										
	Forage									
		Corn silage		24.6	35.3	35.3	-	-	26.0	23.0
		Sorghum silage		-	-	-	37.5	37.5	-	-
		Lucerne silage		-	-	-	-	-	9.0	8.0
		Ryegrass silage		21.1	-	-	-	6.5	9.0	8.0
		Fescue hay		-	2.0	6.0	6.5	-	-	-
	Commercial concentrate mix ^3^			54.3	62.7	58.8	56.0	56.0	56.0	61.0
Dry matter (DM)				43.4	51.8	59.7	55.9	50.4	37.0	41.0
Nutrient										
		NE_L_ (Mcal/kg DM) ^4^		1.63	1.68	1.68	1.68	1.65	1.64	1.64
		Crude Protein		15.9	15.1	16.4	16.5	15.8	14.6	16.7
		Neutral Detergent Fiber		33.1	31.0	28.0	29.5	29.6	36.3	34.4
		Acid Detergent Fiber		16.5	13.6	13.5	13.5	15.1	15.7	15.9
		Ether Extract		3.8	3.5	4.1	3.8	4.6	3.1	3.5
		Starch		22.0	27.0	20.0	18.0	17.0	17.0	18.0

^1^ Autumn calving season. ^2^ Spring Calving Season. ^3^ Based on ground corn grain, wheat bran, soybean meal, sunflower meal, cottonseed meal, canola meal, rumen inert fat, urea, yeast, and minerals. ^4^ Net Energy of lactation, calculated as 1.909 − (0.017 × ADF) according to NRC [34].

**Table 2 animals-13-01211-t002:** Forage management and chemical composition in each season in mixed systems with low (CB-GRZ) or high (OD-GRZ) environmental exposure.

		Herbage		Chemical Composition				
		Allowance ^1^	Mass ^2^	DM	NE_L_ ^3^	CP ^4^	NDF ^4^	ADF ^4^
Autumn 2019	CB-GRZ	16.3	2378	34.7	1.45	9.7	62.0	32.2
	OD-GRZ	15.1	2411	32.4	1.47	13.8	57.9	29.5
Winter 2019	CB-GRZ	17.9	2213	22.7	1.60	19.4	47.9	22.0
	OD-GRZ	19.7	2429	21.7	1.64	18.2	47.8	20.0
Spring 2019	CB-GRZ	23.8	3198	24.0	1.54	14.5	51.5	25.5
	OD-GRZ	23.5	3022	24.5	1.65	16.7	51.2	19.3
Summer 2020	CB-GRZ	26.5	3804	35.7	1.56	14.5	41.0	24.6
	OD-GRZ	26.7	3566	30.3	1.55	17.9	40.1	25.1
Autumn 2020	CB-GRZ	21.1	2497	38.0	1.61	22.4	43.1	21.6
	OD-GRZ	19.8	2337	37.8	1.60	21.3	44.5	22.3

^1^ Expressed as kg DM/cow/day. ^2^ Expressed as kg DM/hectare, estimated at ground level. ^3^ Calculated as (3.2 − 0.028 × ADF) × 0.62 [35]. Expressed as Mcal/kg DM. ^4^ Expressed as % of DM.

### 2.4. Statistical Analysis 

Data were analyzed using the MIXED procedure of SAS (SAS Institute Inc., Cary, NC, USA) with the model: Yij = µ + Ti + WOSj + Bk+ Ti × WOSj + eijk, where Yij is the response variable, Ti is treatment, WOSj is week of study (WOS), Bk is block fixed effect, Ti × WOSj is treatment by WOS interaction effect, and eijk is residual error. Treatments were compared with each other. The cow was considered the experimental unit for milk production and composition, BCS, and metabolite concentrations, while the pen was the experimental unit for TMR and pasture DM intake. Data were analyzed as repeated measures over time. Equidistant distribution was considered for milk production (daily) as well as DM intake (weekly), while uneven distribution was considered for milk solids production, BCS, NEFA, and βOHB. Values at calving were included in the model as co-variables to compare BCS, NEFA, and βOHB between treatments throughout lactation. Each calving season was analyzed separately. Mean comparisons were performed by Tukey-Kramer analysis. Mean differences were considered significant if *p* ≤ 0.05. Results are shown as least square means ± standard error of the mean (SEM). 

## 3. Results

The daily average THI and percentage of the days in which THI was <68, between 68 and 72, and >72, and times of heavy rains (i.e., >50 mm/week during at least 3 consecutive weeks or >80 mm in a week) through the 15 months of both experiments are shown in Figure 1. Table 1 shows the ingredients and chemical composition of the 7 TMR/MR fed in both experiments (ACS and SCS). Table 2 shows HA and HM during each season of the experiment, as well as the chemical composition of all MS.

### 3.1. Autumn Calving Season

Heavy rain occurred during WOS 10, 15, 23, 32 to 35, 41 to 43, and 45. Mild heat waves occurred in WOS 37, 40, and 45, and a severe heat wave occurred at WOS 43 (Figure 1). The MR intake averaged 11.7 ± 3.4 kg DM/cow/day, while pasture was 7.2 vs. 7.3 ± 3.2 kg DM in CB-GRZ and OD-GRZ during periods without total confinement (Table 3). Cows were confined during WOS 6–7, 14–15, and 42–43 due to low HM. During these periods, DMI averaged 21.5 ± 4.3 kg/cow/day.

All milk production response variables were affected by WOS, but none were affected by treatment (Table 4). Milk production had a T × WOS interaction (Figure 2). Although fat and protein content and protein had overall T × WOS interaction effects, there were no specific WOS in which the treatment outcomes differed.

**Table 4 animals-13-01211-t004:** Milk production and composition (means ± SEM) per cow in autumn and spring calving seasons in confined (CB-TMR) and mixed systems with low (CB-GRZ) or high (OD-GRZ) environmental exposure during complete lactation.

			Treatment				*p-*Value		
			CB-TMR	CB-GRZ	OD-GRZ	SEM	TRT	WOS	TRT × WOS
Autumn calving season									
	L/cow/day		-	26.2	26.5	0.16	0.14	<0.01	<0.01
	Fat	%	-	3.55	3.57	0.04	0.74	<0.01	<0.01
		kg/d	-	1.02	1.01	0.02	0.58	<0.01	0.07
	Protein	%	-	3.40	3.44	0.02	0.21	<0.01	<0.01
		kg/d	-	0.98	0.98	0.01	0.80	<0.01	<0.01
	Lactose	%	-	4.82	4.87	0.03	0.28	<0.01	0.06
		kg/d	-	1.42	1.41	0.03	0.73	<0.01	0.14
Spring calving season									
	L/cow/day		35.9 ^a^	27.4 ^b^	26.0 ^c^	0.17	<0.01	<0.01	<0.01
	Fat	%	3.59 ^a^	3.55 ^ab^	3.43 ^b^	0.04	0.03	<0.01	<0.01
		kg/d	1.34 ^a^	1.04 ^b^	1.01 ^b^	0.02	<0.01	<0.01	<0.01
	Protein	%	3.29 ^a^	3.17 ^b^	3.19 ^b^	0.04	0.01	<0.01	<0.01
		kg/d	1.23 ^a^	0.96 ^b^	0.95 ^b^	0.02	<0.01	<0.01	<0.01
	Lactose	%	4.89 ^a^	4.78 ^b^	4.87 ^a^	0.04	<0.01	<0.01	<0.01
		kg/d	1.82 ^a^	1.44 ^b^	1.44 ^b^	0.02	<0.01	<0.01	<0.01

^a,b,c^ Means within season with different superscripts differ (*p* < 0.05). TRT—treatment; WOS—weeks of study; TRT × WOS—interaction.

The BCS patterns during complete lactation adjusted for BCS at calving (3.05 ± 0.27) were affected by WOS for all treatments (Figure 3) but did not differ between treatments (2.67 vs. 2.69 ± 0.02).

Plasma levels of NEFA and βHB were only affected by WOS (Figure 4). The mean levels throughout WOS 2 to 30 were 0.35 ± 0.13 mmol/L of NEFA and 0.72 ± 0.07 mmol/L of βHB.

### 3.2. Spring Calving Season

Heavy rain occurred during WOS 12 to 15, 21 to 23, 25, 30, 34, and 37. Mild heat waves occurred in WOS 17, 20, 25, 28, and 32 to 34, and severe heat waves occurred in WOS 23 and 29 (Figure 1).

The MR intake averaged 12.3 ± 2.3 kg DM/cow/day, while pasture averaged 6.8 vs. 5.8 ± 2.5 kg DM in CB-GRZ and OD-GRZ, not including periods of total confinement (Table 3). Cows were confined during WOS 22–23, 35–38, and 43–44, always as a consequence of low HM, averaging 20.0 ± 1.4 kg DM/cow/day. The totally confined system for SCS cows averaged a daily TMR intake of 25.7 ± 2.8 kg DM/cow/day. 

All production responses were affected by WOS treatment and had a T × WOS interaction (Table 4). Milk production was higher in CB-TMR than MS, and CB-GRZ produced more milk than OD-GRZ. Confinement system cows produced more milk than both MS in all WOS except 4, 36, and 37, whereas CB-TMR equaled CB-GRZ and was higher than OD-GRZ. The MS groups were similar, except in WOS 23, 29, and 37, when CB-GRZ milk production was higher than OD-GRZ. Fat content was higher in CB-TMR than OD-GRZ and intermediate in CB-GRZ. Fat, protein, and lactose yields, as well as protein content, were higher in CB-TMR compared to both MS, which were similar. Lactose content was higher in TMR and OD-GRZ than CB-GRZ. While fat content had a T × WOS interaction in SCS, no treatment differences occurred in any WOS. Protein content was only higher in CB-TMR than OD-GRZ and CB-GRZ in WOS 14. Lactose, protein, and fat yield were higher overall in CB-TMR than OD-GRZ and CB-GRZ.

The BCS was affected by WOS treatment and had a T × WOS interaction (Figure 3). The confined system had a higher BCS than CB-GRZ and OD-GRZ (*p* < 0.01), which did not differ (2.76 vs. 2.62 and 2.59 ± 0.02, respectively). At WOS 15 and 26 onwards, CB-TMR had a higher BCS than MS. During WOS 40, OD-GRZ had a lower BCS than CB-GRZ, which in turn was lower than CB-TMR, with a BCS of 2.67, 2.87, and 3.15 ± 0.05, respectively (*p* < 0.01).

Plasma NEFA was only affected by WOS (*p* < 0.01, Figure 4, mean 0.35 ± 0.03 mmol/L during the first 30 WOS). The plasma βHB was affected by treatment (*p* = 0.01), as well as WOS and the T*WOS interaction (*p* < 0.01). The OD-GRZ had the highest plasma βHB throughout, while the confined system had the lowest mean and CB-GRZ was intermediate (0.91 vs. 0.69 vs. 0.77 ± 0.05 mmol/L, respectively). During WOS 10, OD-GRZ had higher βHB plasma levels than CB-TMR, while CB-GRZ was intermediate (0.91 vs. 0.51 and 0.66, ± 0.15 mmol/L respectively, *p* < 0.01). At WOS 15, OD-GRZ had the highest βHB plasma levels of all WOS and treatments, 1.93 mmol/L, without differences between CB-TMR and CB-GRZ. In WOS 21, OD-GRZ βHB plasma levels declined and were intermediate between CB-GRZ and CB-TMR (0.94, 1.07, and 0.65 mmol/L, respectively, *p* < 0.05).

## 4. Discussion

### 4.1. Autumn Calving Season

In the autumn calving season, the MS subjected to different levels of environmental exposure did not differ from each other in any analyzed variable (i.e., milk production, milk composition, energy metabolism). Although June, October, and December rainfall values exceeded historical averages, during the rest of the year they were below historical values (the cumulative value of the 3 months was 598 vs. 307 mm, and the remaining months were 965 vs. 1369 mm for experimental station and the national historical average, respectively, Instituto Uruguayo de Meteorología) [36]. In addition, heat stress only occurred for 2 weeks at the end of the study at an advanced stage of lactation, when cows are less susceptible to heat stress due to lower intake and production, and therefore lower metabolic heat output [11]. Although the OD-GRZ treatment was more exposed to environmental conditions than the CB-GRZ, good maintenance of the infrastructure in the feeding and resting areas (i.e., cleaning after periods of rain, mound construction), as recommended [37], in addition to correct shade sizing [7] and ad libitum access to fresh water, likely mitigated the negative effects of such exposure. The open-air conditions to which the OD-GRZ treatment cows were subjected were judged to be better than those on commercial Uruguayan dairy farms. In general, commercial dairy farm animal facilities maintenance is less frequent and rigorous than in our study, such that cows face longer and muddier conditions, which are detrimental to their well-being and performance [14,16]. As well, the number of cows per pen is usually much higher than that used in our study, with higher deposition of excreta and higher ground pressure, causing higher moisture and surface deterioration.

The seasonal HM and HA values, as well as paddock access time, indicate that cows had no l imitations [38,39] to reaching forage harvest levels close to 11 kg DM/cow/day [40]. Herbage allowance reached in our MS, characterized by high stocking rates and supplementation, was 38.2 vs. 38.5 for CB-GRZ and OD-GRZ, respectively. It should be taken into account that pasture DM intake values were estimated according to the difference between the energetic contribution of the TMR and estimated production and maintenance requirements, plus a fixed extra maintenance cost of 20% for grazing activity, making it an approximate estimate that allows values to be compared to other studies. Notwithstanding, milk fat content did not differ between groups. In general, increasing pasture inclusion causes a higher acetic:propionic ratio in the rumen [20,41], thereby increasing precursors for de novo milk fat synthesis as opposed to diets with a higher inclusion of concentrate. This result confirms the lack of difference in estimated pasture intake and pasture inclusion diet between our treatments [42].

In ACS, the lack of treatment differences with no T × WOS interaction in BCS is consistent with similar levels of indicators of energetic metabolism (i.e., NEFA and βHB). Calving BCS was at the lower limit of what is considered desirable (3.03 and 3.00 ± 0.27 for both MS) according to Roche et al. [43]. However, cows mobilized tissue to a nadir of 2.58 in WOS 8 and 12, a value that is not critically low, which is supported by NEFA values, which decreased to <0.6 mmol/L in WOS 6, stabilizing in 0.16 ± 0.05 mmol/L between WOS 14 and 30.

### 4.2. Spring Calving Season

In SCS mixed systems, differences occurred in milk production during full lactation among treatments subjected to different levels of environmental exposure. This was mainly due to the cumulative numerical differences between WOS 20 and 37, which was summer. As MR quantity (15.4 ± 3.9 kg DM) and composition, as well as HM and HA, were similar between treatments during this period (3783 ± 2610 kg DM/ha and 28.4 ± 15.9 kg DM/cow, respectively), the lower cow performance in the OD-GRZ treatment (1.5 L less than CB-GRZ) could be due to differences in environmental exposure. It was during this period that the 6 mild waves and the 2 severe heat waves were concentrated, and, during WOS 23, both treatments were in confinement, consuming equal amounts of MR. Thus, it seems that the productive difference was due to differences in environmental exposure, as the first severe heat wave occurred in that WOS. The next significant difference in WOS 29 coincides with the 2nd severe heat wave. At this time, cows were under similar grazing conditions: 16 vs. 18 kg DM/cow of HA and 2150 vs. 2500 kg DM/ha of HM for OD-GRZ and CB-GRZ, respectively. The third significant difference in milk production between MS and WOS 37 was observed. Moreover, during WOS 36 and 37, CB-GRZ milk production did not differ from CB-TMR, but it did from OD-GRZ. Although the conditions to rate WOS 36 as a heat wave were not reached, it averaged a THI of 73.2, which means that cows suffered heat stress. In addition, there was a heavy rain event in WOS 37 that could have affected milk production, as moisture content is aversive to cows’ willingness to lie down. The fact that both MS were in total confinement entails that OD-GRZ cows could not trade off confinement resting behavior deprivation during grazing time [16,18], impairing their welfare and performance.

The higher energy demand due to the activation of adaptation mechanisms [9,10,44], together with the reduction of nutrient absorption and lower udder nutrient uptake [45] cause milk production to decrease by up to 35% [7,13]. Thus, milk production differences between levels of environmental exposure were ~5%, a lower difference that may be due to this being a long-term study that encompasses a full lactation, with part of the experiment occurring in thermoneutral conditions. During summer, the mean difference amounted to 10%, with peak differences of 20% in the weeks where heat waves occurred (i.e., WOS 23 and 29). 

No differences in milk fat and protein content occurred between MS subjected to contrasting environmental exposure, although milk lactose content was higher in OD-GRZ. Smith et al. [46] showed increased milk fat content (3.5 to 3.7%) and decreased milk protein content (3.2 to 3.1%) in heat stressed cows from mild to moderate HS without changes between moderate and severe HS. This is consistent with Wheelock et al. [9], who did not report an effect on milk fat or protein content in heat-stressed cows. In contrast, Cowley et al. [47] observed that HS had a strong influence on milk protein concentration but no effect on that of milk fat or lactose, hypothesizing that HS could provoke specific downregulation of mammary protein synthesis. Quantifying direct physiological effects on milk production is difficult, and inconsistent results have been reported, as it is also strongly affected by behavioral factors that affect nutrient ingestion and absorption, which is even more complex in grazing conditions. The numerically lower pasture DM intake and its inclusion in the diet in OD-GRZ, and therefore higher TMR inclusion, suggests a relatively higher starch passage to the small intestine and higher glucose availability in the mammary gland in comparison to CB-GRZ, which consumed more herbage, which could explain the higher milk lactose content [22,41].

The treatment with the highest environmental exposure expressed an energy metabolism imbalance, evidenced by higher values of βHB between WOS 15 and 19 (~85 to 125 DIM), indicating situations of subclinical ketosis (>1.2 mmol/L [48]), which resulted in lower BCS recovery at the end of the study, compared to treatments less exposed to the environment (CB-GRZ) and/or with a higher feeding level (CB-TMR). Climatic conditions during WOS 12–15 were associated with cold rather than heat stress, as the ambient temperature aver-aged 18.5 °C and heavy rain occurred. Cows subjected to wet surfaces exhibit less lying and a lower quality of rest [16], in addition to the extra energy expended on thermoregulation [49]. The lack of a NEFA surge in OD-GRZ cows coupled to β-hydroxybutyrate may have occurred because of differential tissue utilization of NEFA due to increased physical activity [50,51]; as muscle shivering for thermoregulation [52], less lying and resting time [16], more walking time, and standing in postures that may reduce the amount of surface area exposed to wind and rain [14,49]. These metabolic differences allowed CB-GRZ cows to have different BCS from OD-GRZ cows at WOS 40 of almost a fourth point on the scale of Edmonson et al. [33].

The CB-TMR treatment had 6.4 and 7.1 kg more DM intake than CB-GRZ and OD-GRZ, respectively, which reached pasture inclusion levels of 35.9 and 31.4% of the di-et [27]. In consequence, the confined group exceeded milk production by 31% and 38% compared to the MS of CB-GRZ and OD-GRZ, respectively, and reached an average of 28% more milk solids production than the MS, while maintaining higher and more stable production levels throughout the full lactation compared to systems including grazing. Indeed, MS production levels sharply declined (~31.0 to 23.5 L) from WOS 11 to 18, throughout spring, as daily PA incremented from 18 to 27 kg DM/cow and supplementation decreased from 15 to 11 kg DM/cow. Once summer began, PA diminished again, and therefore supplementation increased, showing a production improvement although not recovering initial values. The early difference in milk production in favor of CB-TMR (~8 L at 30 DIM) leads to the assumption that a short confinement at the beginning of lactation (i.e., the first 3 weeks) with ample TMR supply in conditions of low environmental exposure might be a management alternative in MS in order to maintain high production levels during full lactation.

The higher milk production of CB-TMR vs. MS cows was due in part to higher DM intake, in agreement with other studies that indicate a higher DM intake in more nutrient-rich diets [1,2,42]. Bargo et al. [20] studied the performance of cows in early to mid-lactation during 21 weeks in spring consuming 100% TMR or 32% pasture + 68% MR and housed overnight in a free-stall barn and found that totally confined cows had higher DMI and milk production compared to MS (26.7 vs. 25.2 kg DM and 38.1 vs. 32.0 kg milk/day for each treatment, respectively), which represented an advantage of 19% in milk production during the study. Otherwise, neither milk fat nor true protein content differed between treatments [20]. Similar results were obtained by White et al. [53] in a 4-year study that showed 11% more milk production per lactation in cows fed TMR than MS cows but found no differences between treatments for milk fat or protein content in pasture + con-centrate (occasionally fed with pasture hay or haylage) compared to confined cows fed TMR. Vibart et al. [54] showed that increasing the pasture proportion of the diet caused a quadratic decrease in DM and CP intake, which resulted in lower milk production in cows consuming 35% of the diet as pasture compared to TMR-fed cows (32.7 vs. 36.6 kg/d) dur-ing mid-lactation, housed in free-stall barns in an 8-week study during spring, although there were no differences when consuming pasture during autumn at up to 41% of the diet inclusion of pasture. Again, no differences occurred in milk fat content between feeding systems (3.68 vs. 3.31%) or in CP (2.86 vs. 2.84%) with increased pasture proportion. Salado et al. [24], in a 9-week study in autumn-winter during early lactation, found no change in milk fat (3.88%) or protein (3.43%) content as TMR proportion increased from 25 to 100% of the diet, with gradual increases in productive levels in response to higher DM and energy intake as pasture proportion in the diet decreased. Subsequently, Salado et al. [55] observed higher milk production (33.7 vs. 32.3 kg/d) during early to mid-lactation in TMR fed cows compared to 75% TMR + 25% oat pasture during autumn–winter, with similar milk fat content (3.90%) but higher milk protein content (3.53 vs. 3.47%). In our study, milk protein content was higher in TMR-fed cows than MS-fed cows, in accordance with the latter experiment and with expectations, as DM and energy intake are associated with milk protein content [23]. Contrary to mentioned previous reports, milk fat content was lower in OD-GRZ cows relative to CB-TMR cows, although it was similar between CB-TMR and CB-GRZ cows. High environmental exposure during HS alters ingestive patterns, prompting less frequent and longer meals at a higher intake rate [11], thereby altering rumination patterns and saliva supply to the rumen [56,57]. Finally, fewer ruminal contractions and decreased blood flow to the rumen epithelium in response to HS [58] impair ruminal stability, fermentation, and nutrient absorption [59], which may cause a lower contribution of fatty acids to support the synthesis of milk fat in the mammary gland in OD-GRZ cows. 

In SCS, calving BCS was below recommended values for successful transition to early lactation [43], with treatment means of 2.80, 2.83, and 2.72 ± 0.21 for CB-TMR, CB-GRZ, and OD-GRZ, respectively. For the confined system, the BCS nadir was at WOS 10 (2.58 ± 0.03), while MS continued losing BCS until WOS 16, with a minimum of 2.48 ± 0.03. The confined treatment had a change in BCS slope from WOS 26 onward, which corresponds to 164 DIM, when BCS started recovering, although it was slow compared to that reported for these systems [43], probably due to the low BCS at calving. However, the confined treatment overcame the negative energy balance in less time than the MS, supported by lower levels of βHB in the OD-GRZ cows, and started increasing BCS earlier than in both MS, although there were no differences in plasma NEFA between treatments. The final BCS of MS cows was more than a quarter point lower in the scale [33] than the CB-TMR treatment cows, staying below 3.00 until the end of the study. These results are not consistent with those of Bargo et al. [20], where the final BCS did not differ between CB-TMR and MS, probably because this experiment started at 110 DIM, when the period of greatest lipid mobilization had already occurred. Fajardo et al. [2] did not find differences in BCS when comparing feeding strategies, similar to our results, since the experiment was completed in the 1st 10 weeks of lactation, when cows on both treatments were still in early lactation, as in our study.

## 5. Conclusions

In autumn calving cows, CB-GRZ and OD-GRZ failed to differentiate from each other in any measured response. This was likely due to the lower rainfall compared to the historical average, limited and mild heat stress only at lactation’s end, and good infrastructure design and maintenance in the feeding and resting areas. Spring-calving cows in a fully confined TMR system had the highest milk production and overcame the early lactation negative energy balance in less time than MS. In grazing spring-calving cows, lower milk production and worse indicators of energy metabolism (i.e., higher βHB and lower BCS recovery by the end of the study) showed that heat stress impaired the performance of grazing cows with high environmental exposure. 

Overall, results show that outdoor soil-bedded milk production systems, when well-managed, have a very high milk production potential that could equate to the productive response of improved infrastructure systems (i.e., a compost-bedded pack barn with cooling capacity) under moderately unfavorable environmental conditions, but in worse situations (i.e., the presence of medium or severe heat waves and heavy rain), performance could be compromised.

## Figures and Tables

**Figure 1 animals-13-01211-f001:**
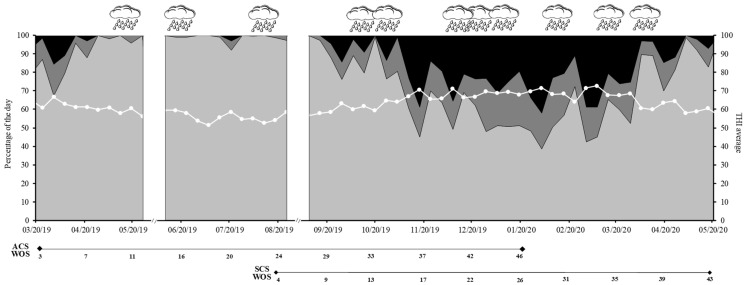
Percentage of the day in which a temperature humidity index (THI) was observed that was less than 68 (
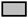
), between 68 and 72 (
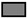
), and higher than 72 (

). The continuous white line with dots indicates the THI weekly average. Cloudy icons indicate moments of heavy rain (>50 mm per week during consecutive weeks or >80 mm in a week). There were no records during part of May and August 2019 due to technical problems at the meteorological station (blank spaces).

**Figure 2 animals-13-01211-f002:**
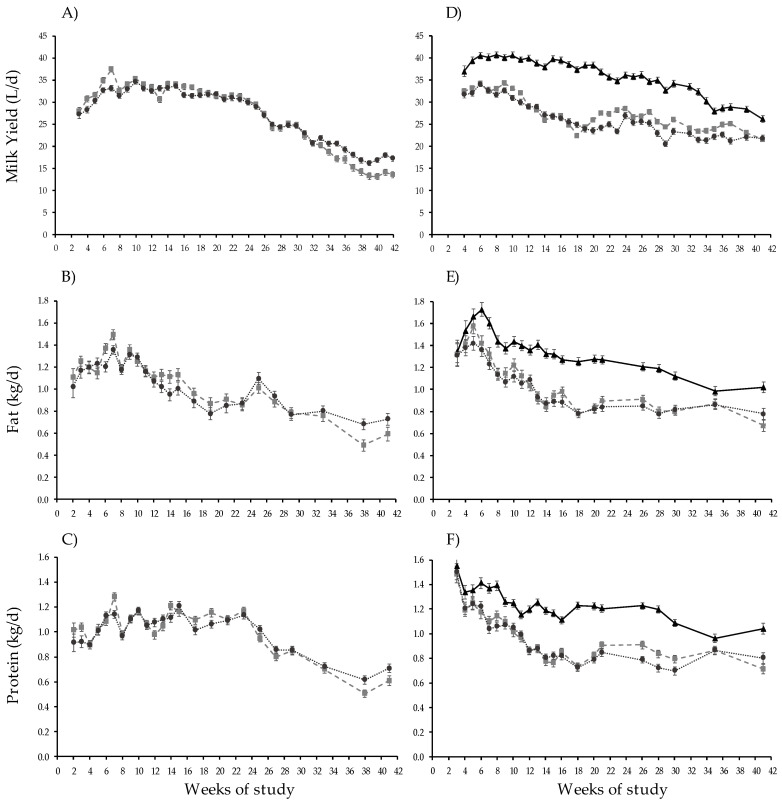
Milk (L/day), fat, and protein (kg/day) yield per cow in autumn (**A**–**C**) and spring (**D**–**F**) calving seasons in confined (
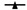
) and mixed systems with low (
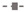
) or high (
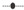
) environmental exposure during complete lactation.

**Figure 3 animals-13-01211-f003:**
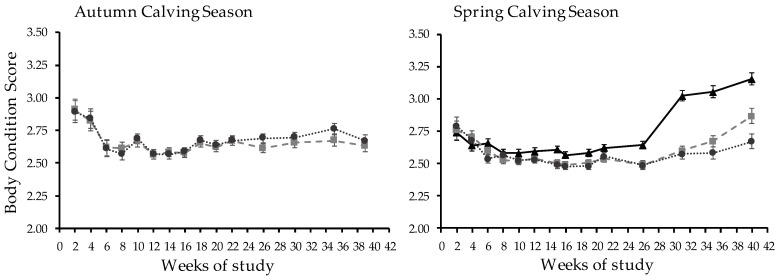
Patterns of body condition score in autumn and spring calving seasons in confined (
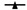
) and mixed systems with low (
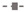
) or high (
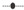
) environmental exposure during complete lactation.

**Figure 4 animals-13-01211-f004:**
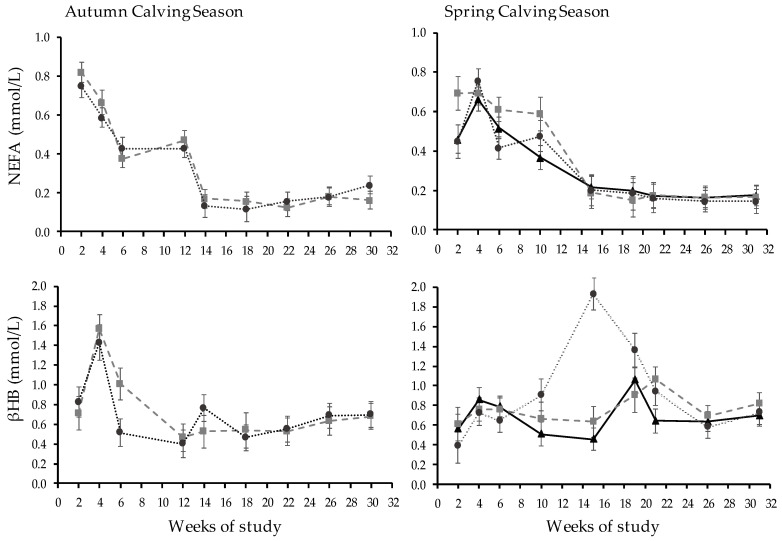
Patterns of non-esterified fatty acids (NEFA, mmol/L) and beta-hydroxybutirate (βHB, mmol/L) plasma levels in autumn and spring calving seasons in confined (
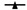
) and mixed systems with low (
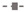
) or high (
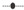
) environmental exposure.

**Table 3 animals-13-01211-t003:** Total mixed ration (TMR) and pasture dry matter intake (DMI) per cow (kg DM/day) in the autumn and spring calving seasons in confined (CB-TMR) and mixed systems with low (CB-GRZ) or high (OD-GRZ) environmental exposure during complete lactation.

		Autmn Calving Season		Spring Calving Season		
		CB-GRZ	OD-GRZ	CB-TMR	CB-GRZ	OD-GRZ
Autumn 2019	Pasture	7.8	7.6	-	-	-
	TMR	14.0	13.6	-	-	-
Winter 2019	Pasture	4.3	4.1	-	5.9	4.7
	TMR	14.5	14.3	24.1	14.0	14.6
Spring 2019	Pasture	9.2	9.3	-	7.2	6.1
	TMR	8.2	8.2	28.0	11.2	11.5
Summer 2020	Pasture	8.8	11.3	-	5.9	5.0
	TMR	8.5	8.5	27.1	12.8	12.8
Autumn 2020	Pasture	-	-	-	9.3	8.9
	TMR	-	-	21.9	10.9	10.9

## Data Availability

Data are available upon reasonable request to the corresponding author.

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
