# Peer review of "Performance of Autumn and Spring Calving Holstein Dairy Cows with Different Levels of Environmental Exposure and Feeding Strategies"

_animals, 2023, doi:10.3390/ani13071211_

Round 1

Reviewer 1 Report

Overall the manuscript is good, but I will suggest to attend the following specific comments. In addition the discussions sections could be improved, discussing the main findings of the present study. 

L26: body condition score may not be an indicator of the energy metabolism but maybe an indirect of the energy balance.

L30: which type of environmental exposure, heat stress???

L34-L36: please rewrite treatments description to make it more clear for the reader.

L36-L45: I will suggest to avoid abbreviation on abstract. Because abbreviation are pretty long, they make information a bit complicate to understand to a reader that is not familiar with the experimental setup.

L49: Please avoid to start the introduction with “Although”, it can be used in the second paragraph but not at first.

L58-L59: please change “in turn”

L64: rewrite “as well as they have longer”, need to include “they” in the sentence.

L65-L69: please rewrite… need to improve the English…

L77-L78: during heat stress (HS), there is an immune system hyperactivated, which demands 1 kg of glucose during around 24h, as such this represents the more energy lost, then after that the energy is used for metabolic adaptation to HS. Please check Baumgart group papers… Rhoads et al., etc.

L79: the low milk and components yield during HS is related to the low energy available for milk and components synthesis, Please check previous reference suggested above…

L82-L84: please rewrite.

L89: replace “facilities” by conditions and include “to heat stress”

L92-L93: why particularities??? Please rewrite the sentence.

L93-L94: I don’t get the point of original, does this mean that nobody does around the south hemisphere?

L106: what do you mean with less environmental exposure? Less time, low temperature? Please specify. I will suggest to reformulate your hypothesis.

L128-L134: please rewrite treatment description.

L173: Herbage biomass

L182: please include reference for THI.

L206-L216: please include references for the calculations.

L217: what does “semi-montly” means?

L218: mL

L220: ‒20°C

Table 1: numbers 1 to 7 do not add extra info to the table.

L265-L266: please provide THI values for mild and severe heat wave… I would like to see in your methodology the reference you used to indicate that you really got severe HS… I know that for Dairy Holstein cows  moderate to severe HS could be > 72 to < 85 TH.

Table 3 and 4: please always write full abbreviation meaning in the bottom of the tables.

Figure 2: a) please your panel need of letter (A-F) in each graphic, then you can explain in the bottom of the figure the season for each of them. b) better Milk yield (L/d), c) figures to the right need Y axis title.

Figure 2-4: I will suggest to include SEM or SD error bar.

Does plasma insulin was assessed? Please provide some discussions of this even if you did not measure… insulin is is a key hormone during HS, it is an antilipolytic hormone. HS cows presents hyperinsulinemia compared to thermoneutral cows.

Reviewer 2 Report

Authors conducted a study comparing how confinement or not could impact on production indexes and blood metabolites in a full-lenght lactaction. It is interesting to have studies like this in the literature to understand the posible impact (or not) of confinement cows performance. It would be welcome to include some stress biomarkers, such as cortisol, to have a deeper understanding of animal welfare.

- ln 82-83. This idea is not accurate at all. Enough space must be provided when cows are confined, thus, space should be enough to lie.

- ln 118-119. Please include THI data to show differences between the two studied periods.

- ln 158 and 173. I would place tables as close as posible to their first reference. It is easier to understand what is referenced in the text about these tables.

- ln 180-181. how did you record the climatic conditions?

- ln 217. what is semi-monthly? please specify

- ln 220. use the appropriate degree symbol

- Results are interesting: in confined situation milk production is statistically higher and BHB lower, in spite of having the same nutrition levels. Do you have any stress indicator? Could it be posible to measure? Something like cortisol or p-substance? It is expectable to find differences, or may be not...

- ln 366-367. I disagree with this comment. We, as veterinarians, should try to maximize this conditions in farms, and we are achieving this target.

- ln 368-370. same comment than before. It is regulated the mininum m2 per cow... we try to maximize everywhere is posible

- spring calving season discussion section is too long. I would shorten.

- do you have any reproduction outcomes? It could be interesting to link this results with its reproduction results or performance throughout lactation
